# Endoplasmic Reticulum Stress and Mitochondrial Stress in Drug-Induced Liver Injury

**DOI:** 10.3390/molecules28073160

**Published:** 2023-04-02

**Authors:** Sisi Pu, Yangyang Pan, Qian Zhang, Ting You, Tao Yue, Yuxing Zhang, Meng Wang

**Affiliations:** College of Veterinary Medicine, Gansu Agricultural University, Lanzhou 730070, China; pssummer0602@163.com (S.P.); panyangyang_2007@126.com (Y.P.); zq880204@126.com (Q.Z.); gsauylyt@163.com (T.Y.);

**Keywords:** ERS, mitochondrial stress, drug-induced liver injury

## Abstract

Drug-induced liver injury (DILI) is a widespread and harmful disease closely linked to mitochondrial and endoplasmic reticulum stress (ERS). Globally, severe drug-induced hepatitis, cirrhosis, and liver cancer are the primary causes of liver-related morbidity and mortality. A hallmark of DILI is ERS and changes in mitochondrial morphology and function, which increase the production of reactive oxygen species (ROS) in a vicious cycle of mutually reinforcing stress responses. Several pathways are maladapted to maintain homeostasis during DILI. Here, we discuss the processes of liver injury caused by several types of drugs that induce hepatocyte stress, focusing primarily on DILI by ERS and mitochondrial stress. Importantly, both ERS and mitochondrial stress are mediated by the overproduction of ROS, destruction of Ca^2+^ homeostasis, and unfolded protein response (UPR). Additionally, we review new pathways and potential pharmacological targets for DILI to highlight new possibilities for DILI treatment and mitigation.

## 1. Introduction

Drug-induced liver injury (DILI) refers to liver toxicity induced by a drug or its metabolite and is commonly categorized into intrinsic and idiosyncratic DILI [1,2]. Non-steroidal anti-inflammatory drugs, anti-tuberculosis drugs, antiepileptic drugs, traditional Chinese medicine (TCM), and other drugs can cause varying degrees of DILI. Due to the complex pathogenic mechanism of DILI, there is no current specific drug. [3,4,5,6,7,8].

DILI is tightly linked to mitochondrial and endoplasmic reticulum stress (ERS). Most proteins are synthesized, processed, and transported in the endoplasmic reticulum, a determinant compartment for ensuring cell homeostasis. Upon stimulation by drugs and their metabolites, the endoplasmic reticulum homeostasis is disturbed, which initiates ERS and can activate apoptosis pathways. This cascade of events potentiates the development of liver injury, causing liver failure in severe cases [9]. Mitochondria provide direct energy for cell function, maintain the environmental balance in hepatocytes, and are the target organelles of oxidative stress injury. Mitochondrial stress, caused by unfolded or misfolded proteins in the mitochondria, alters the permeability of the mitochondrial membrane, which subsequently leads to excessive ATP consumption, the release of reactive oxygen species (ROS), and impaired calcium homeostasis. These events result in mitochondrial swelling and the consequent release of pre-apoptotic factors, leading to hepatocyte apoptosis [10,11].

Communication between the ERS and mitochondrial stress is also an important topic of investigation. When cells undergo mitochondrial stress, the redox balance in the endoplasmic reticulum is disrupted, interfering with the endoplasmic reticulum function and triggering ERS [12]. Xiao et al. showed that drug-induced ERS could stimulate the expression of the mitochondrial stress factor HSP60, impair mitochondrial respiration, and decrease mitochondrial membrane potential in mouse hepatocytes [13]. This proves that ERS can also induce mitochondrial stress.

DILI has a high prevalence worldwide; however, improvements in the therapeutic approaches for this event are still lacking. Recent studies have identified new potential targets for the treatment of DILI, such as sphingosine kinase-1 (SPHK1) [14], steroidogenic acute regulatory protein (STARD1) [15], transient receptor potential 2 (TRPM2) [16], and activating transcription factor 4 (ATF4) [17]; genes associated with ERS and mitochondrial stress. Notably, inhibition of the SPHK1–STARD pathway effectively alleviates DILI. In this review, we discussed the mechanisms of liver injury caused by several types of drugs, focusing primarily on DILI caused by ERS and mitochondrial stress. Importantly, we highlight the main cellular events that mediate ERS and mitochondrial stress, such as the overproduction of ROS, impaired Ca^2+^ homeostasis, and UPR (unfolded protein response). Additionally, we highlight new (and old) pharmacological targets, aiming to contribute to the development of DILI treatment and mitigation.

## 2. Classification of Drugs Producing DILI and Their Associated Mechanisms in the Endoplasmic Reticulum and Mitochondrial Stress

### 2.1. DILI Classification According to the Liver Lesion and Drug Type

Generally, DILI is divided into intrinsic and idiosyncratic types, but it can also be classified according to the lesion and drug. The pathological targets of DILI involve hepatocyte injury (67.3%), bile duct epithelium damage (23.9%), and vascular injury (8.8%) [18]. Hepatocyte injury includes mitochondrial damage, calcium imbalance, and excessive ROS production, the majority of which occur due to lobular hepatitis; bile duct epithelial damage, manifestations of cholestatic patterns, a hepatocellular injury, vascular injury, including the obliteration of portal venules, and veno-occlusive disease or sinusoidal obstruction syndrome may also occur. According to the developmental processes of DILI, mild to severe diseases involve mild liver enzyme elevation, jaundice, hepatic steatosis, hepatitis, liver fibrosis, cirrhosis, hepatic portal vein sclerosis, liver failure, angiosarcoma, liver cancer, and death [19]. Mild diseases, if not treated timely, may deteriorate to severe forms or cause death [20].

The drugs that can cause DILI can be divided into antibiotics, non-steroidal anti-inflammatory drugs, antipsychotics, antidepressants, lipid-lowering drugs, anti-tuberculosis drugs, antiepileptic drugs, traditional Chinese medicine (TCM), dietary supplements, oral anticoagulants, anti-androgen drugs, antibacterial drugs, hypoglycemic drugs, and acid-suppressing drugs. The extension and type of lesions differ with each drug (Table 1). Nonetheless, some mechanisms involved in DILI development with certain drugs remain unclear. Currently, the drugs known to induce ERS and mitochondrial stress include acetaminophen (APAP), isoniazid (INH), and Valproic acid (VPA) [5,8,21,22].

### 2.2. Endoplasmic Reticulum and Mitochondrial Stress in Drug-Induced Liver Injury

#### 2.2.1. ERS and ERS-Related Molecules in the Development of DILI

The endoplasmic reticulum (ER) comprises a nuclear membrane domain, smooth ER, rough ER, and regions of connection with other organelles. Peptide chains are processed and transported through the ER to the Golgi apparatus [56]. After the reticular Golgi apparatus processes and sorts the peptide chains, the resulting protein with a native conformation is expelled into the extracellular space. One of the main functions of the ER is to store calcium, which is released upon various signaling events in a controlled manner. The subsequent increase in cytoplasmic Ca^2+^ results in signal transduction [57]. The ER encompasses a quality control system that deals with excessive protein accumulation, protein misfolding, or continuous loss of Ca^2+^, which ensures cell death upon failure to adapt to a stressor.

When misfolded proteins accumulate, cells can degrade them via ER-associated protein degradation (ERAD) [58]. The ERAD involves the cytoplasmic ATPase p97, through which a membrane reverse-transcription translocation complex provides a pathway for the protein to return to the cytoplasm. Dislocated ERAD substrates are ubiquitinated at the cytoplasmic side of the ER membrane and targeted for proteasomal degradation [9]. When the ER cannot deal with the high load of protein folding, Endoplasmic reticulum stress (ERS) ensues, triggering the UPR, which enhances the protein processing ability of the ER. The UPR helps refold misfolded proteins in the ER, allowing its return to homeostasis. If the stress signal is severe or maintained, ER initiates the apoptotic pathway to remove the misfolded proteins and restore homeostasis [59].

APAP-induced endoplasmic reticulum stress

After APAP enters the liver, 90% is converted into non-toxic compounds under the influence of UGTs (glucuronidase) and SULTs (sulfur transferase) and excreted in the urine. The remaining 10% is metabolized by the cytochrome P450 (CYP450) enzyme system to form the metabolite N-acetyl-p-benzoquinonimine (NAPQI). In the normal (therapeutic doses) condition, NAPQI is detoxified with glutathione (GSH) [60]. However, when APAP ingestion reaches a toxic dose, a large amount of APAP enters the CYP450 pathway, and NAPQI is produced in larger amounts [49] (Figure 1). Once GSH is depleted, NAPQI binds to cellular proteins, such as mitochondrial protein, which initiates hepatocellular necrosis and DNA fragmentation [61]. Additionally, when GSH content decreases, the ER lumen produces a redox shift, protein disulfide isomerase (PDI) undergoes oxidation, and the redox imbalance damages the function of ER oxidoreductase, which may initiate the ER-related signaling pathway [62,63], This event activates endoplasmic reticulum-dependent signal transduction and apoptosis, which may be a leading mechanism of ERS [64]. Recent reports suggest that ERS and ERS-related molecules participate in developing APAP-induced liver injury. Nagy et al. showed significant induction of an ERS-responsive proapoptotic transcription factor, GADD153/CHOP (growth arrest and DNA damage-inducible gene 153 or C/EBP-homologous Protein), in a mouse model of APAP-induced liver injury [65]. CHOP participates in APAP-induced ERS during liver injury [66]. A recent report also suggested that upregulation of a mitochondrial cholesterol transporter, steroidogenic acute regulatory protein (STARD1), following ERS mediates APAP-induced hepatoxicity via mitochondrial SH3BP5 (SAB) and JNK1 (c-jun N-terminal kinase 1) and 2 phosphorylation [67]. Li et al. found that SPHK1 levels in the liver significantly increased after APAP treatment. Phosphorylation of SPHK1 can catalyze the formation of S1P (Sphingosine-1-phosphate) (Figure 2), which can activate PERK-eIF2α-ATF4 and ATF6 (Activating Transcription Factor 6) and induce the generation of the apoptosis signal CHOP, prompting ERS [14]. These experimental findings suggest that ERS and its related molecular events occur in liver injury via the regulation of chemical chaperons. Wang et al. demonstrated that INH-induced liver injury shares the same mechanisms as that of APAP, primarily by ONOO^−^ generated by oxidative stress.

#### 2.2.2. Mitochondrial Stress and Mitochondrial Stress-Related Molecules Participate in the Development of DILI

Mitochondria consist of a mitochondrial inner membrane (IMM), mitochondrial membrane gap, mitochondrial outer membrane (OMM), and the mitochondrial matrix. In cells, mitochondria are the main source of NADH and are responsible for parts of the pyrimidine and lipid biosynthesis pathways, including the fatty acid β-oxidation pathway [68]. Similar to the ER, unfolded or misfolded proteins in the mitochondria increase mitochondrial pressure, resulting in mitochondrial stress. It was shown that loss of mitochondrial membrane potential, decreased canonical fusion proteins, and alterations in mitochondrial lipid composition can drive changes in mitochondrial morphology and that these changes lead to mitochondrial fragmentation and reduced function [69]. Mitochondrial dysfunction is associated with several human diseases, such as metabolic syndrome, cancer, and neurodegenerative diseases. Mitochondrial dysfunction also plays a key role in the pathogenesis of DILI, a consequence of altered metabolic pathways and damaged mitochondrial components [10,11]. When the mitochondrial death critical threshold is surpassed, mitochondrial damage can trigger liver necrosis or failure leading to the activation of cell death signaling pathways [10,70]. More specifically, sodium valproate impairs mitochondrial respiration, aspirin affects mitochondrial β-oxidation, and diclofenac damages the mitochondrial membrane.

APAP-induced mitochondrial stress

At higher APAP doses, mitochondrial shape has a biphasic response. The early changes in mitochondrial morphology are reversible and help pre-serve mitochondrial function. In contrast, late delayed change in mitochondrial morphology is irreversible, and these later changes tilt the scales toward mitochondrial fission, leading to mitochondrial fragmentation and reduced functionality [69]. Excess APAP-generated NAPQI interferes with the electron transport chain, when complex III is responsible for leaking electrons, which come into contact with oxygen and generate superoxide radicals [71]. The produced ROS activate the JNK pathway, and phosphorylated JNK is transferred to the mitochondria and binds to the Sab protein, further aggravating oxidative stress and ROS production [72]. Superoxide radicals are converted into hydrogen peroxide (H_2_O_2_) and molecular oxygen (O^2−^) by manganese superoxide dismutase (MnSOD) in the mitochondria or produce peroxynitrite (ONOO^−^) when combined with NO. GSH or antioxidant enzymes (such as catalase) can scavenge H_2_O_2_. Excessive free radicals cause GSH depletion, which results in the accumulation of ONOO^−^ in the mitochondria, eventually leading to mitochondrial DNA damage and the formation of nitrotyrosine protein adducts. The mitochondrial damage caused by adduct formation triggers the collapse of the mitochondrial membrane potential under redox conditions and hepatotoxicity, resulting in mitochondrial stress and liver damage [73]. Active JNK induces changes in the mitochondrial membrane permeability, leading to mitochondrial swelling and necrosis, culminating in liver cell damage [70]. Mitochondrial oxidative stress initiates extracellular death signaling, causing extracellular proteins (such as Bax) to migrate to the mitochondria. Bax forms pores on the outer membrane of the mitochondria, leading to the release of intermembrane proteins [60]. These proteins are then transferred to the nucleus, resulting in DNA breakage. In addition, mitochondrial oxidative stress and peroxynitrite cause MPT (mitochondrial permeability transition) through the MPTP (mitochondrial permeability transition pore). MPT can trigger different cellular responses, from the physiological regulation of mitophagy to the activation of apoptosis or necrosis [74].

Isoniazid-induced mitochondrial stress

Although INH and APAP share pathogenic mechanisms in liver injury, the hepatotoxicity of INH is more severe than that of APAP [8]. In INH-induced DILI, isoniazid is catalyzed by CYP2E1, generating the toxic metabolites acetylhydrazine and hydrazine. The covalent binding of acetylhydrazine to biomacromolecules in hepatocytes causes liver injury. The former can hydrolyze into hepatotoxic hydrazine, leading to the depletion of GSH, resulting in changes in ERS and mitochondrial membrane permeability (Figure 1). Furthermore, hydrazine forms superoxide by inhibiting mitochondrial complex II, promoting mitochondrial stress and liver injury [75]. It can also directly cause mitochondrial damage owing to the formation of isoniazid or its metabolites. For example, INH reduces ATP production in mitochondria by blocking the electron flow, which promotes oxidative stress and energy homeostasis imbalance in the mitochondria [76].

Valproic acid induced mitochondrial stress

Valproic acid (VPA) is an anti-seizure drug that causes idiosyncratic liver injury [77]. The VPA metabolite, 2-propyl-4-pentenoic acid, has been implicated in VPA-induced hepatotoxicity [78]. Furthermore, VPA can deplete inositol, which increases the expression of fatty acid elongases, catalyzing the synthesis of fatty acids and ceramides. Increased ceramide levels decrease the expression of amino acid transporters, which induces cellular stress due to nutritional stress and ERS induction [21]. VPA and its metabolites interfere with mitochondrial function mainly through the upregulation of the fatty acid transporter (CD36) and inhibition of β-oxidation, which increases fatty acid content, decreases ATP and NADH, and inhibits the mitochondrial respiratory chain. This further increases ROS, leads to cytochrome c release, activates caspases, and results in mitochondrial stress, ultimately causing hepatocyte apoptosis and liver injury [41]. VPA also upregulates progesterone, which promotes an accumulation of cholesterol in the mitochondria and the structural alteration of the mitochondrial surface transporters, ultimately inducing mitochondrial stress [22].

## 3. ERS Triggers and Signaling Pathways in DILI

Several endogenous and exogenous factors can affect ER function and cause ERS. Exogenous factors include oxygen deficit, radiation, toxic chemicals, and infection by pathogenic microorganisms [57,79]. Endogenous factors include abnormal calcium regulation, lipid metabolism disorders, etc. They are described in detail below because DILI mainly involves endogenous factors.

### 3.1. ERS Triggers

#### 3.1.1. Abnormal Calcium Regulation

The thapsigargin-induced ERS involves the inhibition of sarco/endoplasmic reticulum Ca^2+^-ATPase (SERCA), which results in severe calcium depletion in the ER. In addition to the typical inducers of ERS, the rise in free cytosolic calcium also induces apoptosis in different types of cells treated with thapsigargin, which induces ERS by inhibiting the SERCA, resulting in serious consumption of ER calcium [80]. In addition to the activation of downstream effectors that trigger cell death induced by ERS, an increase in free cytoplasmic calcium is also an effective pro-apoptotic signal in different types of cells. Trimetazidine, matrine, lonomycin, and other drugs can trigger ERS by disrupting normal calcium metabolism.

#### 3.1.2. Lipid Metabolism Disorders

The liver plays a central role in lipid homeostasis; under normal physiological conditions, the lipid input is equal to the lipid output of the body. The interruption of this balance promotes lipid metabolism disorders. Rituximab, an HIV protease inhibitor (HPI), can cause lipid metabolism disorder by increasing the accumulation of free cholesterol that consumes the ER calcium pool [81], increases SREBP activity to activate UPR, induces apoptosis, and promotes the formation of foam cells [82].

### 3.2. Endoplasmic Reticulum Stress Signaling Pathways Involved in DILI

#### 3.2.1. Three UPR Pathways

Misfolded proteins are prone to toxic aggregation; therefore, eukaryotes have evolved a UPR to ensure the normal dynamic balance of protein folding. The UPR maintains protein folding in the ER, thereby inhibiting the toxicity associated with the accumulation of unfolded proteins [83]. This process occurs under the regulation of immunoglobulin heavy chain binding protein (Bip, GRP78), an amino-terminal chaperone of the endoplasmic lumen end. An increase in unfolded protein promotes the dissociation of Bip, releasing its inhibitory effects over three transmembrane proteins and initiating the stress response. These three proteins, ATF6, inositol-requiring enzyme-1α (IRE1α), and PKR-like ER kinase (PERK), help the ER recover from stress [84].

First, activated ATF6 enters the Golgi and hydrolyzes the S1P and S2P (Sphingosine-1-phosphate) enzymes. Subsequently, ATF6 dissociates from the dictyosome and enters the nucleus as a transcription factor binding to the ERSE (ERS-response element), inducing the transcription of multiple genes, including Bip, CHOP, and X-box binding protein 1 (XBP1) (Figure 3) [85].

Second, the IRE1 pathway mediates the transcriptional induction of ER quality control (including molecular chaperones, folding enzymes, and endoplasmic reticulum-associated degradation components) and secretory proteins. IRE1α activates its nuclease activity via homodimerization and autophosphorylation [86]. IRE1α splices XBP1 mRNA to obtain the transcription factor XBP1, which is transferred into the nucleus to regulate UPR and ERAD-related genes and alleviate the damage caused by the ERS environment. However, sustained and severe ERS promotes apoptotic signaling through the IRE-1 pathway. Activated IRE1α can recruit TRAF2 (TNF receptor-associated factor 2) and with ASK1, phosphorylate JNK and NF-κB (Figure 4). JNK phosphorylates several Bcl-2 family members and promotes cytochrome c release, caspase activation, and apoptosis. Thus, signals initiated from the cytoplasmic kinase domain of IRE1α are primarily pro-apoptotic signals involving the JNK pathway [87]. The NF-κB pathway is involved in apoptosis, cell survival, and the activation of inflammatory cells. Continuous activation of this pathway can lead to uncontrolled cell growth and increased levels of tumor growth (such as IL-1β, TNF, and IL-6) and anti-apoptotic factors. This pathway can also activate autophagy in a MAPK-dependent manner, eventually leading to hepatic stellate cell (HSC) activation and liver fibrosis. However, ERS promotes the apoptosis of activated HSC, reducing and rescuing liver fibrosis [88]. Liver pathological manifestation of chronic liver diseases, such as non-alcoholic liver injury, chronic hepatitis B, and chronic hepatitis C. If not treated actively, damaged hepatocytes can stimulate the proliferation of intrahepatic fibrous connective tissue, progressing from liver fibrosis to cirrhosis.

Third, Bip release activates the PERK pathway that promotes the phosphorylation of eIF2α, triggering an integrated stress response (ISR) that inhibits protein translation and reduces protein load in the endoplasmic reticulum [89]. This phenomenon increases the expression of ATF4 (Activating Transcription Factor 4), which enters the nucleus to regulate the expression of UPR target genes. Under the condition of ERS transition, PERK promotes eIF2α phosphorylation and induces ATF4 translation expression through homologous dimerization and autophosphorylation self-activation. Eventually, the transcription of the ERS apoptosis marker protein CHOP is promoted, culminating in the activation of the apoptosis signaling pathway (Figure 5) [90].

Following ERS induction, the body restores cell homeostasis through the UPR, and the final fate of the cells is determined by several factors. You et al. found that the UPR response transcription factor, QRICH1, determines whether cells move towards adaptation or programmed death during the final UPR period. As a regulator of a unique transcription module, QRICH1 coordinates cellular stress responses to regulate protein synthesis and secretion under steady state and pathological conditions [91]. The consequences of ERS also depend on the duration and intensity of the stress [92].

#### 3.2.2. Caspase-12 Pathway

After ERS, the caspase-12 pathway is activated. ERS induces high CHOP expression, which promotes hepatocyte apoptosis by downregulating the expression of the apoptosis gene Bcl-2 combined with the UPR-related molecules. ERS causes a calcium imbalance in the ER and activates caspase-12, which subsequently activates caspase-9 and caspase-3 by inducing an increase in molecular chaperones, such as Bip and GRP94, ultimately inducing hepatocyte apoptosis. Under steady-state conditions, TRAF2 forms a stable complex with caspase-12, and endoplasmic reticulum stress induces caspase-12 to separate from TRAF2 and promote its dimerization (Figure 6) [93].

## 4. Mitochondrial Stress Signaling Channels Activated by Drug-Induced Liver Injury

Integrated stress response (ISR) is an evolutionarily conserved intracellular signaling network activated in response to internal and external stresses. ISR is induced by four eIF2α kinases, namely PERK, amino acid general control non-inhibitory 2 (GCN2, which senses amino acid reduction), double-stranded RNA-dependent protein kinase (PKR, which senses viruses), and eIF2α kinase (HRI, which senses heme deficiency). UPR in the ER mainly includes the IRE1α, ATF6, and PERK pathways. The PERK-eIF2α-ATF4 pathway is a branch of the ISR; mitochondrial stress can cause eIF2α phosphorylation through ISR, inhibiting overall protein translation while upregulating the translation of proteins, such as ATF4 and ATF5, which initiate the mitochondrial UPR (UPRmt) [94]. Two mitochondrial stress conduction pathways exist in mammals: the ATF4-ATF5-CHOP and the OMA1–DELE1–HRI pathways [95].

The first mitochondrial stress signal channel is the ATF4-ATF5-CHOP. Mitochondrial dysfunction stimulates the phosphorylation of eIF2α, which translates to the transcription factors CHOP, ATF4, and ATF5. The ATF5 activity is negatively regulated by mitochondrial import into healthy mitochondria. Under steady-state conditions, ATF5 is introduced into mitochondria and degraded by LONP1. Under mitochondrial stress, CHOP, ATF4, and ATF5 are transferred to the nucleus, and UPRmt occurs. ISR stimulates the upstream open reading frame (uORF)-regulated translation of CHOP, ATF4, and ATF5 proteins, which subsequently activate transcription, reconnect cell metabolism, and enhance mitochondrial protein homeostasis. Mammals When the mitochondrial stress time exceeds a certain threshold, mitochondria are cleared through mitotic phagocytosis [83].

The second stress signal channel is the OMA1–DELE1–HRI pathway. Kampmann et al. found that mitochondrial stress is transmitted to the cytoplasm through the OMA1–DELE1–HRI pathway [96]. Mitochondrial stress stimulates OMA1 (a mitochondrial stress-activated protease)-dependent cleavage of DELE1, which promotes its accumulation in the cytosol in the form of DELE1s. The cytosolic DELE1s bind to HRI, activate eIF2α kinase activity, phosphorylate eIF2α, and upregulate downstream ATF4 translation, triggering ISR to ensure mitochondrial homeostasis [95]. Phosphorylation of eIF2α leads to a decrease in overall protein levels but an increase in the transcription factors ATF4, ATF5, and CHOP (Figure 7). Basic leucine zipper (bZIP) regulates the upregulation of transcription factor ATF4 translation that binds to DNA targets to promote cell survival and maintain homeostasis. However, when stress is severe or when its duration exceeds a certain threshold, the ISR signal is converted to regulate apoptosis and clear damaged mitochondria [97]. Additionally, Nour et al. found that HRI is essential for the downstream signaling of NOD1 and NOD2, which can induce NF-κB signaling [98], activate inflammatory cells, and cause inflammation in the liver.

## 5. Communication between Mitochondrial and Endoplasmic Reticulum Stress

### 5.1. Calcium Exchange

#### 5.1.1. Contact Site between ER and Mitochondria-MAM

The ER and mitochondria are in physical contact through a dynamic membrane, the mitochondria-associated membrane (MAM). The MAM contains cholesterol and sphingolipids for increased thickness [99,100] and an intracellular lipid raft-like domain that is closely apposed to the mitochondria, both physically and biochemically. The ER and mitochondria undergo calcium ion exchange, energy exchange, and lipid transport through the MAM. MAM can maintain the steady state of both organelles through contact between the ER and mitochondrial membrane. Therefore, the operation of normal functions of the MAM plays an important role in ensuring the steady state of the human body. Studies have shown that MAM-localized proteins can regulate the UPR. PERK is a key controller of oxygen and phosphorus; although it is not located in the mitochondria, it can be found in the MAM. When the concentration of cytosolic Ca^2+^ increases, dimerized PERK interacts with silk protein A, rearranges the protein cytoskeleton, and increases the MC (mitochondrial complex) formation. This interaction promotes the formation of endoplasmic reticulum-plasma membrane contacts [101]. IRE1α has also been shown in MAM; furthermore, the interaction of IP3R3 determines the structure and the Ca^2+^ signaling on the MAM [102].

#### 5.1.2. ER Receptor-Mediated Ca^2+^ Exchange

The sigma-1 receptor (Sig-1R) is a resident protein of the ER that localizes to the MAM. Deletion of Sig-1R can damage ER-mitochondrial contact sites and affect intracellular calcium signal transduction and mitochondrial morphology, accompanied by ERS and mitochondrial defects [103]. IP3R (inositol 1,4,5-triphosphate receptor) releases calcium from the endoplasmic reticulum and increases the intracellular calcium concentration. Sig-1R forms a complex to stabilize IP3R with the ER partner, Bip, on the MAM; however, when Ca^2+^ in the ER is depleted or stimulated by ligands, Sig-1R dissociates from Bip, leading to the long-term signal transduction of Ca^2+^ into the mitochondria through IP3R. Excessive ERS leads to an imbalance in protein homeostasis. The release of calcium ions from the ER into the cytoplasm leads to the opening of the mitochondrial permeability transition pores, forming a positive feedback regulation and bidirectional apoptosis [104]. As mentioned previously, calcium imbalance also activates caspase-12 and the production of pro-inflammatory cytokines, such as IL-1β and IL-18, a key step in ERS-induced apoptosis. Calcium imbalance also affects mitochondrial-induced apoptosis by coupling Bip on the MAM with voltage-dependent anion channel protein 1 (VDAC1) (Figure 8) [71].

#### 5.1.3. TRPM2 Involved in Ca^2+^ Exchange

Transient receptor potential 2 (TRPM2) is a nonselective cation channel located on the cell membrane. Several intracellular signaling pathways mediate ROS-induced hepatocyte injury; however, one of the main events is the ROS-induced increase in Ca^2+^ influx and the subsequent increase in cytoplasmic Ca^2+^ concentration [105,106,107]. The TRPM2 channel allows Ca^2+^ and Na^+^ to flow through the plasma membrane and release Ca^2+^ and Zn^2+^ from lysosomes. In different animal cell types, calcium mediates ROS-induced cell damage and death through the TRPM2 channel. Under oxidative stress, ROS activates poly ADPR polymerase (PARP), which induces ADPR production and TRPM2 channel activation, increasing calcium influx [108]. TRPM2 transport from the intracellular membranes to the plasma membrane contributes to the activation of TRPM2 in APAP-induced hepatocytes [16]. The activation of the TRPM2/Ca^2+^/CaMKII signaling pathway promotes lipid accumulation, mitochondrial damage, and ERS, which aggravates the progression of non-alcoholic fatty liver disease (NAFLD) to non-alcoholic steatohepatitis (NASH), then to cirrhosis and ultimately, to liver cancer [109]. Overall, calcium homeostasis in intracellular mitochondria and the endoplasmic reticulum plays a determinant role in maintaining human health.

### 5.2. Relationship between Mitochondrial and Endoplasmic Reticulum Stress

Xiao et al. found that chemical-induced ERS can stimulate the expression of the mitochondrial stress protein HSP60, impair mitochondrial respiration, and reduce mitochondrial membrane potential in mouse hepatocytes. Moreover, HSP60 overexpression induces ERS by increasing Bip and CHOP levels. HSP60 regulates ERS-induced liver lipid production through the mTORC1-SREBP1 signaling pathway [13]. ERS can cause eIF2α phosphorylation through ISR, inhibit protein translation, and upregulate the translation of proteins such as ATF4 to initiate mitochondrial stress [83,89]. These reports suggest that mitochondrial stress is a consequence of endoplasmic reticulum stress.

Research shows that a large amount of ROS is produced upon mitochondrial stress, and when the redox balance in the endoplasmic reticulum is disrupted, which triggers ERS [72]. Therefore, it appears that ERS is also a consequence of mitochondrial dysfunction. ERO1α-PDI (protein disulfide isomerase) in ER catalyzes the formation of disulfide bonds in proteins, transferring electrons to oxygen to produce H_2_O_2_ [110]. This produces a large amount of ROS and releases a large amount of Ca^2+^ into the cytoplasm, which results in the opening of the MPTP and mitochondrial swelling, thereby aggravating mitochondrial oxidative stress. Therefore, swelling of mitochondria and ERS do not exist independently in severe DILI, mutually affecting each other through various channels to determine liver cell damage.

## 6. The Immunological Mechanisms of APAP-Induced Liver Injury

In the immune response induced by hepatotoxicity-mediated APAP liver injury, Kupffer cells form the first line of defense by recognizing necrotic hepatocytes through danger-associated molecular pattern molecules (DAMPs) and pathogen-associated molecular patterns (PAMPs). Kupffer cells release cytokines and chemokines upon activation, leading to acute inflammation marked by an increase in acute phase proteins [111]. Concomitantly, cytokines amplify the inflammatory process by promoting the release of other inflammatory mediators from infiltrating leukocytes, which upregulate adhesion molecules (for example, ICAM-1 and CD11b/CD18), regulating immune cell aggregation through mediator production. The expression of adhesion molecules also contributes to neutrophil adhesion and translocation within the sinusoids and dependent oxidative stress that promotes hepatocyte death [112,113]. Additionally, activated B cells produce antibodies participating in APAP-induced liver injury. Although several studies have shown that the immune response plays an important role in DILI [1,43,114], its association with ERS and mitochondrial stress requires investigation.

## 7. Pathways for the Treatment of a Drug-Induced Liver Injury

### 7.1. SPHK1–STARD1 Pathway

SPHK1 deficiency leads to a decrease in S1P content, inhibiting MPT, ATF4, and ATF6 levels, and inflammatory gene expression. SPHK1 plays an important role in hepatotoxicity induced by ERS and mitochondrial stress. Using SPHK1 antagonists and gene knockouts in mice treated with APAP, Li et al. found that ERS was alleviated in mice, MPT was inhibited, expression of inflammatory factors was suppressed, and the risk of hepatotoxicity was reduced [14]. Therefore, targeting SPHK1 may be a novel strategy for the treatment of drug hepatotoxicity.

In mouse studies, Torres et al. found that after ERS occurs, the body mediates the upregulation of STARD1 through the phosphorylation of SH3BP5, JNK1, and JNK2 [67], which results in increased mitochondrial cholesterol content and hepatotoxicity. Knockout of STARD1 can prevent cholesterol accumulation in mitochondria, thereby reducing the risk of hepatotoxicity [15]. Therefore, targeting STARD1 may provide a novel strategy for hepatotoxicity treatment, but effective inhibitors are yet to be developed.

Knockout of SPHK1 alleviates mitochondrial and ERS, whereas the knockout of STARD1 only can alleviate ERS. Inhibition of the SPHK1–STARD1 pathway can reduce DILI through the alleviation of organelle stress. Whether the knockout of SPHK1 affects STARD1 or the knockout of STARD1 affects SPHK1 expression needs to be determined to identify the upstream trigger of the SPHK1–STARD1 pathway. Therefore, studying the mechanism of this pathway and its inhibitors is important as it might prove to be a novel strategy for treating drug-induced hepatotoxicity.

### 7.2. ATF4-ATF5-CHOP Pathway

ATF4 is the main mediator of ISR; Koumenis et al. showed ATF4 upregulation in every tumor studied [115]. After ATF4 knockout, tumor cells die from an excessive internal stress response [17] and exhibit reduced angiogenesis without causing significant damage to mice. The Koumenis Lab is currently working on the development of ATF4 inhibitors. Considering its role in DILI pathogenesis, ATF4 is a potential target for DILI treatment.

ATF5 has tumor cell type heterogeneity and plays a role in promoting tumorigenesis and deterioration in several tumors. ATF5 expression is reduced in liver cancer tissues and acts as a tumor suppressor gene [116]. CHOP is a marker of ERS, and its inhibition can effectively reduce the consequences of ERS [117]. The commonly used methods to investigate this pathway include siRNA-targeted silencing and shRNA-targeted interference, CHOP antibodies, and CHOP inhibitors. Both ATF5 and CHOP might be pivotal targets for DILI therapeutic approaches.

### 7.3. TRPM2 Pathway

Zhang et al. found that the knockdown of TRPM2 alleviated injury through the activation of autophagy and inhibition of the NLRP3 inflammasome pathway [118]. In TRPM2 knockout mice, the APAP-induced liver injury, assessed through the blood concentration of liver enzymes and histology, was significantly decreased compared with that in wild-type mice [119]. This finding suggests TRPM2 is a therapeutic target for APAP-induced liver injury.

Curcumin is a natural antioxidant product that prevents ROS-induced liver injury by inhibition of TRPM2. Even at nanomolar concentrations, curcumin inhibits Ca^2+^ entry into hepatocytes through the TRPM2 channels [120]. This phenomenon shows that the TRPM2 channel is a potential pharmacological target for the prevention of ROS-induced liver injury. Professor Zhang Liang Ren from Peking University developed a class of TRPM2 channel inhibitors that are structural derivatives of anthranilic acid. Broadly speaking, A23 is a highly active and selective TRPM2 inhibitor [121].

### 7.4. JNK Signaling Pathway

The processing and folding of proteins in the endoplasmic reticulum cause ERS, which activates JNK. There are several compounds used to inhibit JNK signaling. The most used is SP600125 [122], which can reduce liver mitochondrial oxidized glutathione (GSSG) levels while promoting liver GSH recovery, revealing a protective effect mainly due to the inhibition of mitochondrial oxidative stress [123]. Leflunomide (LEF) is a commonly used antirheumatic drug in clinical practice that protects hepatocytes from APAP-induced liver injury. LEF can inhibit the phosphorylation of JNK1 and JNK2, preventing changes in mitochondrial permeability and the release of pro-apoptotic factors, thereby protecting liver cells from damage [124]. BI-78D3 is another JNK inhibitor that effectively prevents CCl4-induced acute liver injury [125]. Additionally, ASK1 is a MAP3 kinase that activates the downstream terminal kinases, JNK and p38. The ASK1 inhibitor, GS-444217, ameliorated NASH and improved fibrosis in preclinical studies and a short-term clinical trial [126]; however, this inhibitor was not effective in late-phase clinical trials [127,128]. Currently, biologics and the transplantation of stem cells have been shown to inhibit JNK in DILI through modulation of GSH resynthesis [129].

### 7.5. NF-κB Pathway

Intracellular and extracellular stimuli bind tumor necrosis factor receptor-associated factor (TRAF) to the receptor interacting protein (RIP). RIP proximal signal adaptor protein activates the IKB kinase complex (IKK). IKK activates IKB kinase, which phosphorylates and ubiquitinates IKB protein, degrades, and releases the NF-κB dimer. Finally, NF-κB is activated, translocates to the nucleus, and promotes transcription of target genes. NF-κB is a central link between liver injury, fibrosis, and hepatocellular carcinoma and represents a target for the prevention or treatment of DILIs [130]. Some common inhibitors associated with NF-κB channels are listed in Table 2.

### 7.6. UPR Pathway

Here we describe that PERK, IRE1, and ATF6 can induce apoptosis and activate inflammatory pathways by mediating the phosphorylation of eIF2α, which can cause different degrees of liver damage. The stress response can be reduced by inhibiting the expression of these three proteins. Several available compounds are currently used to regulate UPR, but some challenges and limitations to their use remain. In addition, harm can be reduced by inducing the generation of Bip. The main inhibitors are listed in Table 2. In addition to inhibitors, the CRISPR/Cas9 system can also be used for UPR pathway gene knockout.

## 8. Conclusions

DILI is a complex disease modulated by numerous metabolic, genetic, and environmental mechanisms. Mitochondrial stress and ERS are key factors in DILI, such that liver injury is both a cause and a consequence of these events, creating a positive feedback loop that may promote the development and progression of hepatic injury. Here, we propose that the SPHK1–STARD1 pathway may play a role in determining the outcomes of drug-induced liver damage. This concept requires further experimental testing to identify which other targets might be used to therapeutically prevent or delay DILI progression. Over the last decade, a significant effort has been invested in targeting signaling proteins involved in organelle stress and an array of inhibitors is now available to be tested in the clinic. However, these molecules show limitations, off-target effects being a case in point. A further challenge to be addressed in the future will be to develop tool compounds into agents that can safely be administered in the clinic.

## Figures and Tables

**Figure 1 molecules-28-03160-f001:**
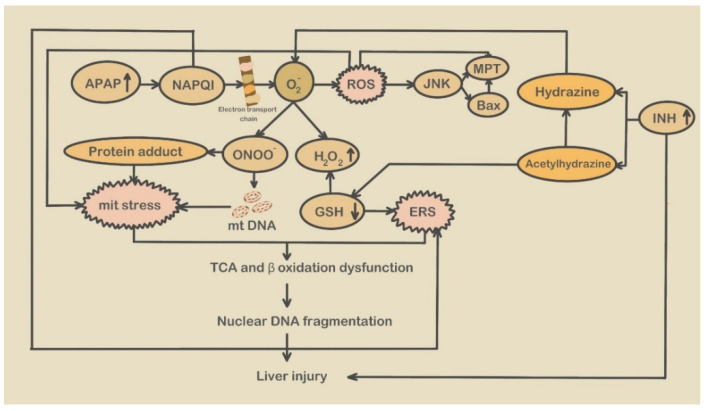
Mitochondrial stress and ERS in APAP and INH-induced liver injury. (APAP: Acetaminophen; NAPQI: N-acetyl-p-benzoquinone imine; ROS: Radical Oxygen Species; JNK: c-jun N-terminal kinase; MPT: Mitochondrial phosphate transporter; Bax: BCL2-Associated X; INH: Isoniazid; mit stress: Mitochondrial stress; mt DNA: Mitochondrial Deoxyribonucleic acid; GSH: Glutathione; ERS: Endoplasmic reticulum stress; TCA: Tricarboxylic acid).

**Figure 2 molecules-28-03160-f002:**
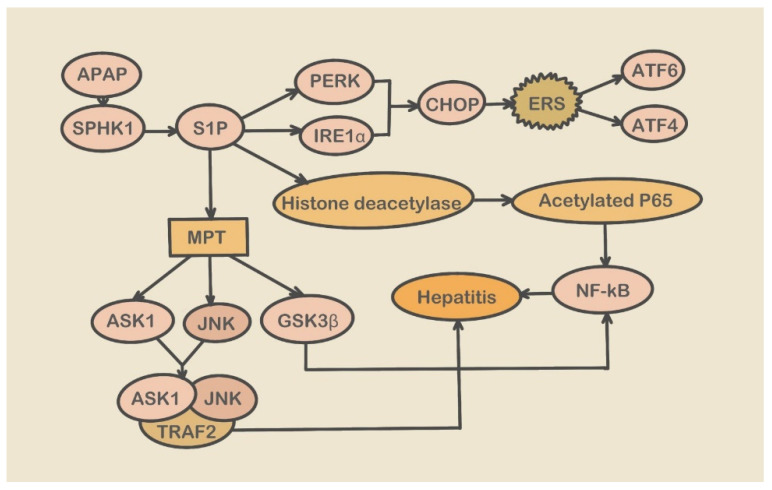
SPHK1 pathway in liver injury. (SPHK1: sphingosine kinase-1; S1P: Sphingosine-1-phosphate; PERK: PKR-like ER kinase; IRE1α; Inositol-requiring enzyme-1α; CHOP: C/EBP-homologous Protein; ATF4: Activating Transcription Factor 4; ATF6: Activating Transcription Factor 6; ASK1: Apoptosis signal-regulating kinase 1; GSK3β: Glycogen Synthetase Kinase 3β; TRAF2: TNF receptor-associated factor 2; NF-κB: Nuclear Factor Kappa B).

**Figure 3 molecules-28-03160-f003:**
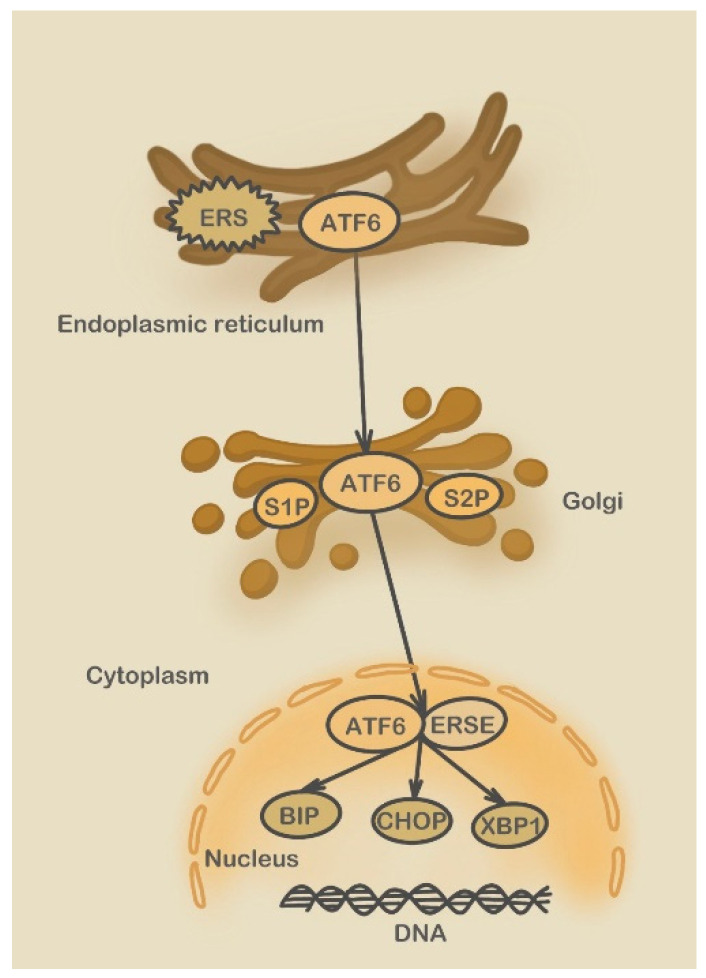
ATF6 pathway. (S2P: Sphingosine-1-phosphate; ERSE: ERS-response element; BIP: binding protein; XBP1: X-box binding protein 1).

**Figure 4 molecules-28-03160-f004:**
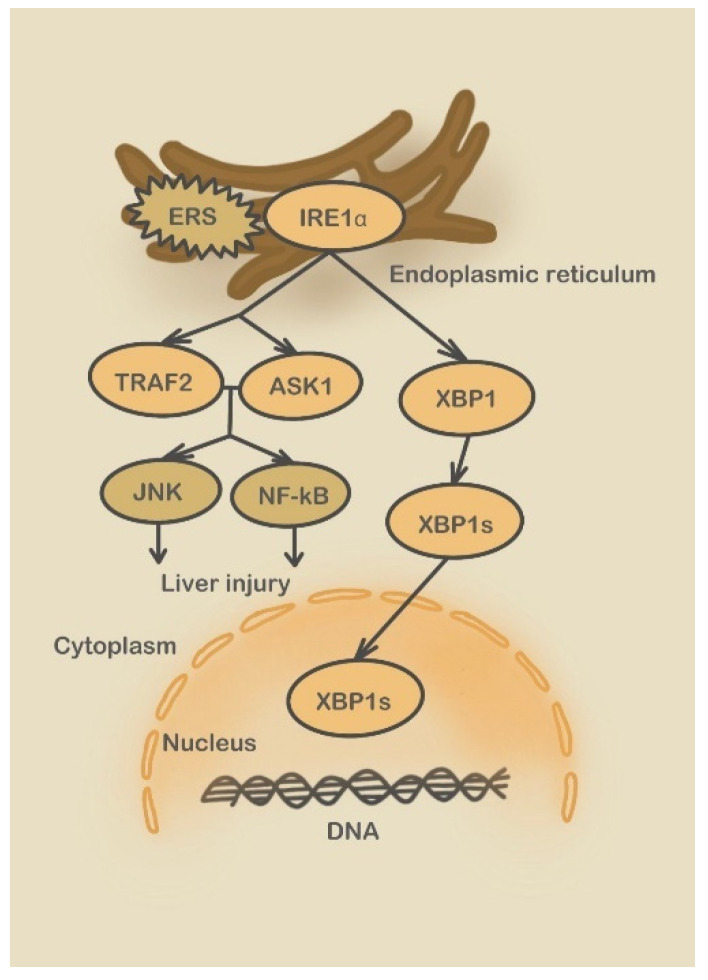
IRE1 pathway. (S2P: Sphingosine-1-phosphate; ERSE: ERS-response element; BIP: binding protein; XBP1: X-box binding protein 1).

**Figure 5 molecules-28-03160-f005:**
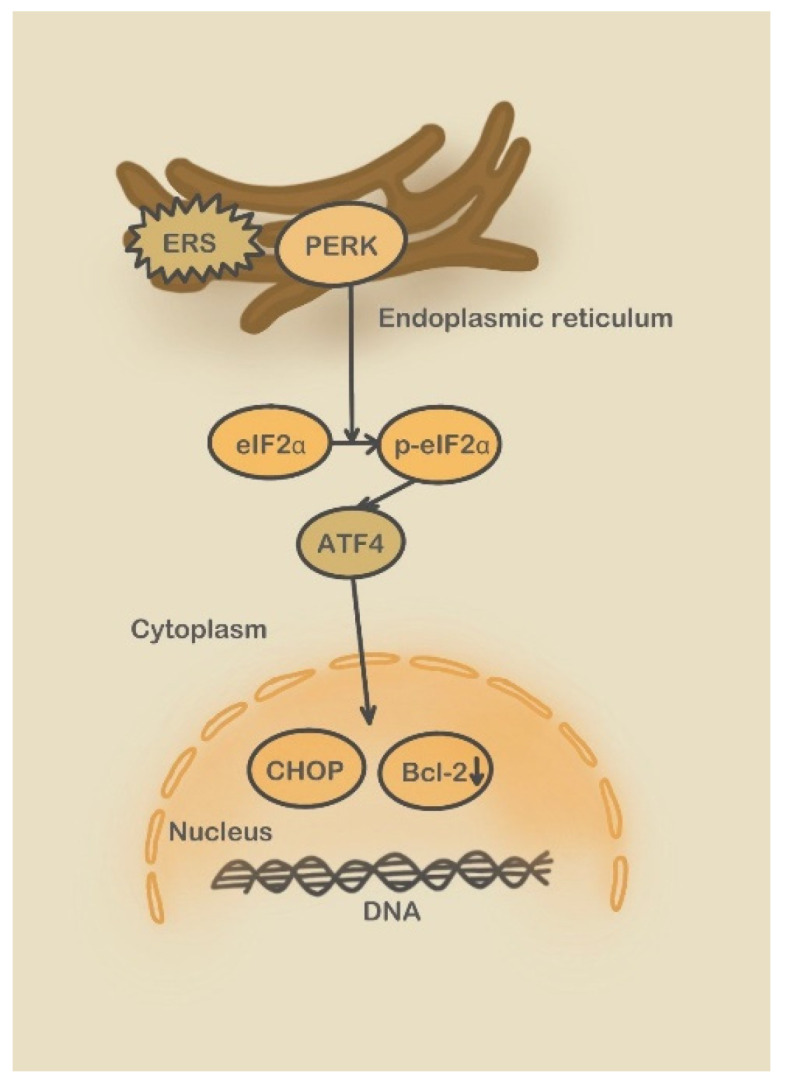
PERK pathway.

**Figure 6 molecules-28-03160-f006:**
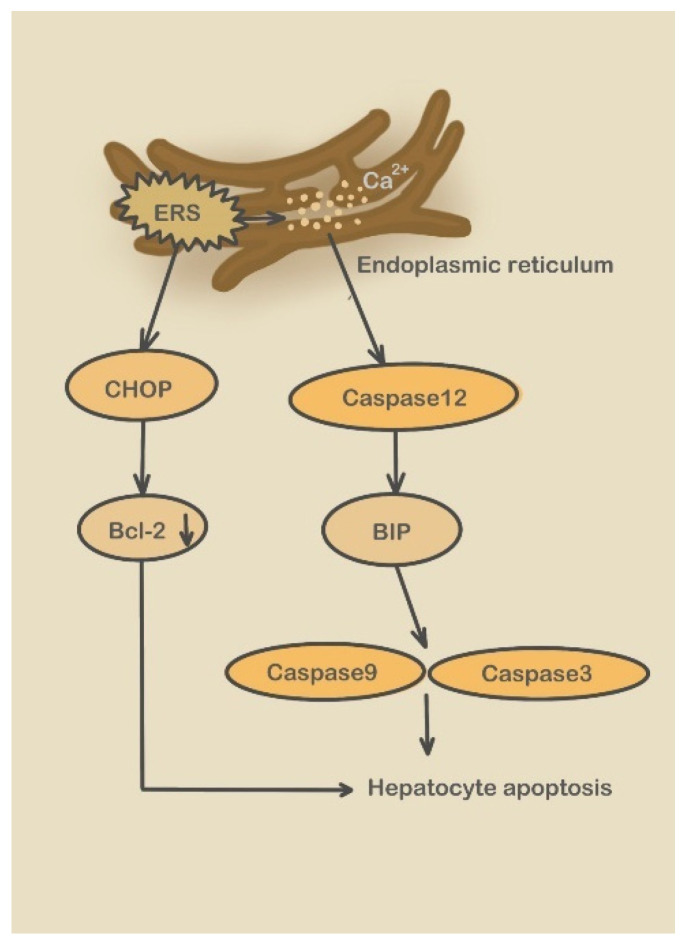
Caspase-12 pathway.

**Figure 7 molecules-28-03160-f007:**
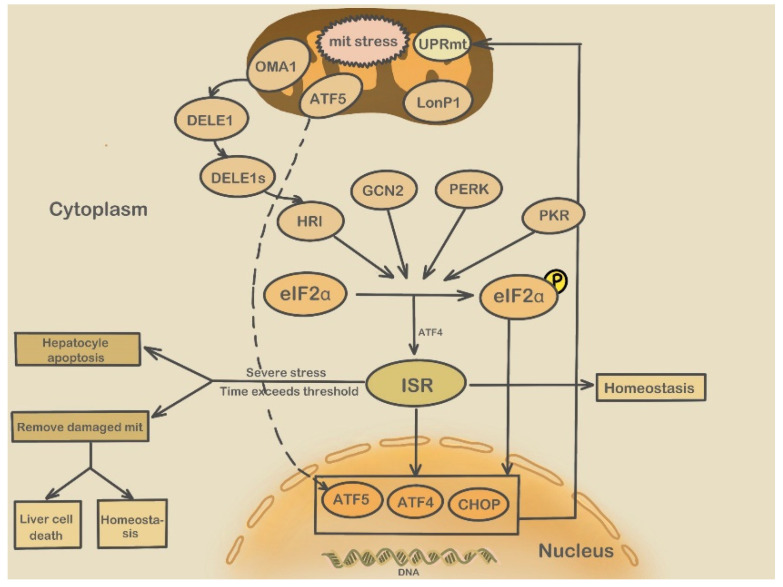
Mitochondrial stress pathways.

**Figure 8 molecules-28-03160-f008:**
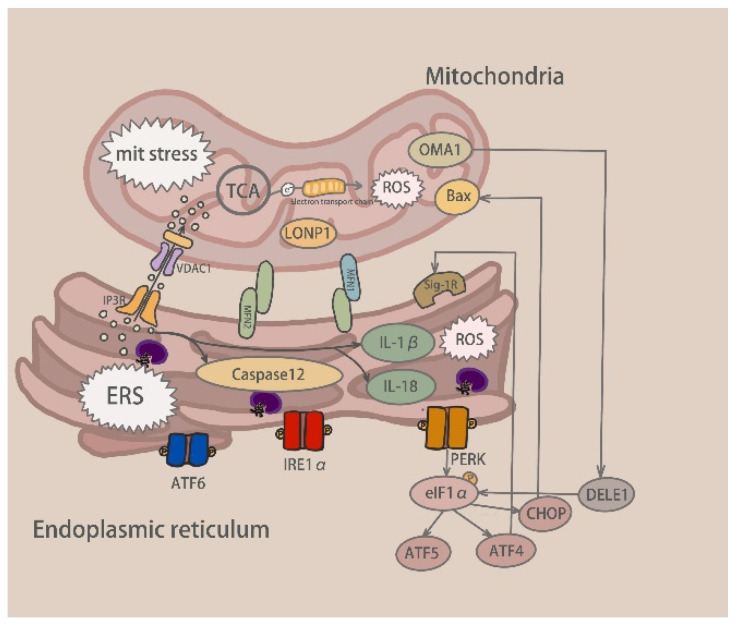
Communication between the ERS and mitochondria stress (LONP1: Lon peptidase1; VDAC1: Voltage dependent anion channel protein 1; IP3R: Inositol 1,4,5 triphosphate receptor; MFN1: Mitofusin 1; MFN2: Mitofusin 2; Sig-1R: Sigma 1 receptor; DELE1: DAP3 binding cell death enhancer 1).

**Table 1 molecules-28-03160-t001:** Common drugs causing liver injury.

Drug Class	Compound	Drug Target	Damage Type	References
Antibiotics	Clindamycin	Hepatocyte; Bile duct epithelial	Hepatitis; Hepatic failure	[23]
	Oxytetracycline	Hepatocyte; Vascular injury	Fatty degeneration	[24]
	Cefazolin	Hepatocyte	Hepatitis	[25,26]
	Nitrofurantoin	Hepatocyte	Hepatitis; Jaundice; Necrosis; Fibrosis	[25]
Non-steroidal anti-inflammatory drugs	Acetaminophen	Hepatocyte	Hepatonecrosis	[4,5]
	Ibuprofen	Hepatocyte; Bile duct epithelial; Vascular injury	Hepatocyte damage; Hepatic failure; Acute hepatitis	[27,28]
	Aspirin	Hepatocyte; Bile duct epithelial; Vascular injury	Hepatic enzymes elevations; Jaundice; Acute hepatitis	[28]
	Diclofenac	Hepatocyte	Hepatitis; Jaundice; Hepatauxe	[29]
Antipsychotics	Risperidone	Hepatocyte; Bile duct epithelial	Hepatitis	[30,31]
	Clozapine	Hepatocyte; Bile duct epithelial	Fulminant liver failure	[32]
Antidepressant	Amitriptyline	Hepatocyte; Bile duct epithelial	Hepatitis; Fulminant liver failure or death	[31]
	Trazodone			
	Agomelatine			
	Bupropion			
	Iproniazid			
	Duloxetine			
	Imipramine			
	Nefazodone			
	Tianeptine			
	Phenelzine			
Lipid-lowering drugs	Atorvastatin	Hepatocyte; Bile duct epithelial; Vascular injury	Hepatitis; Jaundice	[33]
Chemotherapeutics	Fluorouracil	Hepatocyte	Hepatitis; Steatosis	[34,35]
	Irinotecan	Hepatocyte	Hepatitis; Steatosis	[34]
	Oxaliplatin	Hepatocyte; Vascular injury	Hepatic sinusoidal obstruction syndrome	[36]
Antituberculosis drugs	Isoniazid	Hepatocyte; Bile duct epithelial	Hepatitis; Necrosis	[6,7,8]
	Rifampicin	Hepatocyte; Bile duct epithelial	Hepatitis; Necrosis	[6,7,37]
	Pyrazinamide (PZA)	Hepatocyte; Vascular injury	Hepatitis; Necrosis	[38,39]
Antiepileptic drugs	Lamotrigine	Hepatocyte	Hepatic enzymes elevations; Death; Necrosis	[40]
	Valproic acid (VPA)	Hepatocyte	Hepatitis;	[41,42]
Traditional Chinese medicine (TCM)	HeShouWu	Hepatocyte	Hepatitis; Jaundice; Necrosis	[43]
	JuSanQi	Hepatocyte; Vascular injury	Hepatitis; Hepatic sinusoidal obstruction syndrome	[3]
	Dioscorea bulbifera L.	Hepatocyte; Bile duct epithelial; Vascular injury	Hepatitis	[44]
	CangErZi	Hepatocyte; Vascular injury	Hepatitis	[45]
Oral anticoagulants	Rivaroxaban	Hepatocyte; Bile duct epithelial; Vascular injury	Hepatitis; Hepatic enzymes elevations; Jaundice	[46,47]
	Apixaban	Hepatocyte; Bile duct epithelial;	Hepatitis	[48,49]
Antiandrogen drug	Flutamide	Hepatocyte	Hepatitis; Jaundice; ALF	[50,51]
Antimicrobial	Ketoconazole	Hepatocyte	Liver cirrhosis; Jaundice	[52]
Hypoglycemic agents	Troglitazone *	Hepatocyte	ALF	[53]
Acid-inhibitory drugs	Esomeprazole	Hepatocyte	Hepatic enzymes elevations	[54]
	Cimetidine	Hepatocyte	Jaundice; ALF	[55]

* Troglitazone has been discontinued due to its severe hepatotoxicity. When applying such drugs, liver function should be closely monitored.

**Table 2 molecules-28-03160-t002:** Pathway inhibitors in DILI.

Pathway	Class	Compound	Comments	Target	References
SPHK1-STARD pathway	SPHK1 inhibitors	PF543	Sphingosine kinase inhibitor	SPHK1	[14,131]
		SK1-I	Enhances autophagy and cancer cell death	SPHK1	[132,133]
		LCL351	Reduces the expression of pro-inflammatory cytokine	SPHK1	[134]
		SKI-178/349	High selectivity and low toxicity	ATP- binding site	[135]
		SK-F	Without significant systemic toxicity	SPHK1	[136]
		SLC4011540	High cell permeability	SPHK1	[137]
		11b	Selective inhibition of SPHK1	SPHK1	[138]
		CHJ01	Anti-inflammatory	SPHK1	[139]
		SK1-II	Up-regulates Cer level	SPHK1	[140]
		DMS	Causes severe hemolysis in mice	SPHK1	[141]
	S1P inhibitors	NIBR0213	Reduces peripheral blood lymphocyte counts	S1P1	[142]
		JTE013	Inhibits inflammasome priming and inflammatory cytokine	S1P2	[143]
		FTY720	Reduces and inhibits T cells	S1P3	[144]
TRPM2 pathway	TRPM2 inhibitors	Curcumin	Prevents ROS-induced liver injury	TRPM2	[120]
		Compound A23	Highly active and selective	TRPM2	[121]
JNK pathway	JNK inhibitor	SP600125	Inhibits mitochondrial stress	JNK1/2/3	[123]
		LEF	Inhibits JNK1/2 phosphorylation	JNK1/2	[124]
		BI-78D3	Binds D-domain of JIP1	D-domain of JIP1	[125]
	ASK1 inhibitor	GS-444217	Noneffective in late-phase clinical trials	ASK1	[126]
NF-κB pathway	IKK inhibitors	Parthenolide	Specific IkB inhibitor	IkB	[145]
	NF-κB inhibitors	SN50	NF-κB inhibitory peptides for permeable cells	NF-κB	[146]
		BAY11-7082	Prevents IkBα phosphorylation	NF-κB	[147]
	Immunosuppressive agents	PG490	Target gene transcription inhibitor	P65	[148]
	Protease inhibitors	MG-132	Inhibits phosphorylation of IkBα	IkBα	[149]
	Antioxidants	ALLN	Prevents IkBα degradation	IkBα	[150]
		PDTC	Prevents IkB release	IkB	[151]
UPR pathway	PERK inhibitors	GSK2606414	RIPK inhibitor; KIT inhibitor	EIF2AK3	[152,153]
		GSK2656157	RIPK inhibitor	ATP- binding site	[154,155]
		PERK-IN-2	Low renal clearance	ATP- binding site	[152]
		PERK-IN-3	High renal clearance	ATP- binding site	[152]
		AMG PERK 44	GCN2 inhibitor	PERK	[156]
	IRE1 inhibitors	Compound 3	Compound 3 disfavors IRE1 oligomerization	STING	[157,158]
		Compound 15	EGFR inhibitor	EGFR	[157,158]
	ATF6 inhibitors	PF-429242	Inhibits SREBP signaling	S1P	[159]
		Ceapin	Does not affect ABCD3 function	Tethers ATF6 cytosolic domain and ABCD3	[160]
	Bip inducer	BIX	ERS inhibitor	Bip	[161]

## Data Availability

All data presented in this study are available on request from the corresponding authors.

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
