# Peer review of "Endoplasmic Reticulum Stress and Mitochondrial Stress in Drug-Induced Liver Injury"

_molecules, 2023, doi:10.3390/molecules28073160_

Round 1
Reviewer 1 Report
The manuscript “Endoplasmic reticulum stress and mitochondrial stress in drug-induced liver injury.” by Pu et al. is aimed at summarizing the signaling pathways involved in the endoplasmic reticulum (ER) stress and mitochondrial stress, with relation to drug induced liver injury (DILI).
This is an extensive review of recent pathways involved in ER and mitochondrial stress and the possible role of DILI drugs acetaminophen, isoniazid and Valproic acid on these processes. However, the manuscript needs a careful attention of the choice of words in the text as this can change meaning. A careful revision of the English is warranted. Also there are some formatting issues and typos (examples: the subtitles on line 88 and 134 are the same.…use of the word proliferation for fibrosis….typos in words such as caspase in both text and figure..., the description of abbreviations needs to be first time mentioned in the text…etc)
Reviewer 2 Report
The tables provide a nice summary of the text and serve as valuable reference. Specific comments to address are listed below:
Introduction:
§ “DILI, which results in substantial cellular and tissue damage, however, the treatment of DILI is in the exploratory stage” – needs to be reworded.
§ “Recent studies have identified new possible targets for the treatment of DILI…” – need to add appropriate references.
Section 2.1:
§ “Liver cells also undergo injury via apoptosis and necrosis” – liver cells do not “undergo injury via apoptosis and necrosis” these cells may die by apoptosis or necrosis but the injury is due to the topic of the review (e.g. mitochondrial damage/stress, DNA damage, membrane damage etc.)
§ Line 75-77: “…may deteriorate into cirrhosis, liver cancer, and so on” – suggest to say “may deteriorate into cirrhosis and liver cancer.” – what else would be “so on” – lacks precision for a scientific review.
§ Line 82: What other drugs, besides the ones listed, are encompassed by “etc” – either provide an exhaustive list of drugs or be specific and have a reason to mention them (e.g. you will proceed to discuss them in your review).
Section 2.2.1:
§ Line 90-95 missing references. Need to provide references.
§ “When bound GSH is depleted…” – needs to be reworded, awkward phrasing makes its sound like GSH is “bound” to something. GSH detoxifies NAPQI because GSH is nucleophilic, and NAPQI is a reactive electrophile which results in GSH oxidation into GSSG.
Section 2.2.2:
§ “NAPQI also interferes with mitochondrial electron transport chain, allowing electrons to come into contact with oxygen to produce superoxide radicals” – should also discuss more recent evidence elucidating which ETC complex is responsible for leaking electrons which leads to JNK activation (PMID: 33290831).
Section 4.1:
§ Line 312 “mitochondria are mainly composed of….” Reword to “Mitochondria consist of a mitochondrial inner membrane…” etc.
§ In this section, should briefly discuss the relationship between mitochondria structure and function, and how mitochondrial dynamics play a role as an adaptive mechanism to DILI (PMID: 33432343).
Section 5.1.1:
§ Line 377: “According to research…” entire sentence needs to be rewritten (probably safe to just remove completely).
Section 6.3:
- While there is discussion of one of the early JNK inhibitors SP600, this discussion could be expanded beyond small molecules, increasing the novelty and timeliness of this review. For example, biologics and the transplantation of stem cells have been shown to inhibit JNK in DILI through modulation of GSH resynthesis (PMID: 36057886).
Reviewer 3 Report
This is a typical paper, and one of several on DILI that have been published in the recent years. However, after revision it can be considered for publication.
Below, please find comment and suggestions:
Title of table 1 should be adopted correctly. Now it is “Table 1. This is a table. Tables should be placed in the main text near to the first time they are cited.”
Table 1 – pathological target ??? – Should rather be drug target
Table 1 - use horizontal lines to separate types of drug groups
Table 1 – Terramycin – please use inn
Table 1 – Troglitazone – is now not in clinical use (due to hepatotoxicity) – mark it with * and in the subscript of the table provide relevant information
In Figures – legend should include all abbreviations expanded
The structure of chapters should be rearranged: e.g. 2.2. Mitochondrial stress and endoplasmic reticulum stress in APAP, INH and VPA and then you have 4. Mitochondrial stress in drug-induced liver injury or 5. Communication between the endoplasmic reticulum stress and mitochondrial stress
Chapter 2 should rather be entitled – Drugs producing DILI not like it is - 2. Classification of drug-induced liver injury and typical drugs causing endoplasmic 61 reticulum and mitochondrial stress
2.1. Types of DILI according to lesion and drug –
As the last mechanisms discussed chapter (move it right above current chapter 6) I would suggest to place chapter - 2.2. Mitochondrial stress and endoplasmic reticulum stress in APAP, INH and VPA, as an example of drugs description DILI based on previously presented mechanisms in chapters 3, 4 and 5.
The chapter on Immunological Mechanisms of DILI is missing
Round 2
Reviewer 2 Report
The authors significantly improved the manuscript.
Reviewer 3 Report
Accept in present form